# The Impact of International Nonproprietary Names Integration on Prescribing Reimbursement Medicines for Arterial Hypertension and Analysis of Medication Errors in Latvia

**DOI:** 10.3390/ijerph191610156

**Published:** 2022-08-16

**Authors:** Anna Gavrilova, Maksims Zolovs, Gustavs Latkovskis, Inga Urtāne

**Affiliations:** 1Department of Pharmaceutical Chemistry, Faculty of Pharmacy, Riga Stradins University, LV-1007 Riga, Latvia; 2Red Cross Medical College, Riga Stradins University, LV-1009 Riga, Latvia; 3Statistical Unit, Faculty of Medicine, Riga Stradins University, LV-1048 Riga, Latvia; 4Institute of Life Sciences and Technology, Daugavpils University, LV-5401 Daugavpils, Latvia; 5Institute of Cardiology and Regenerative Medicine, University of Latvia, LV-1586 Riga, Latvia; 6Latvian Center of Cardiology, Pauls Stradins Clinical University Hospital, LV-1002 Riga, Latvia

**Keywords:** eHealth, fixed-dose combination, FDC, electronic prescription, database, pharmacists, dispensing, regulatory changes, primary care

## Abstract

The use of international nonproprietary names (INNs) has been mandatory for prescriptions of state-reimbursed drugs in Latvia since 1 April 2020. In a retrospective analysis, we aimed to examine the impact of the new regulation on changes in the prescribing and dispensing practice of antihypertensive agents with an example of bisoprolol or/and perindopril and their combinations. All state-reimbursed bisoprolol and/or perindopril prescriptions for arterial hypertension were evaluated in two time periods: 1 April 2018 to 31 March 2019 and 1 April 2020 to 31 March 2021. The proportion of INN prescriptions increased from 2.1% to 92.3% (*p* < 0.001, φ = 0.903). The rate of fixed-dose combinations (FDCs) increased from 60.8% to 66.5% (*p* < 0.001, φ = 0.059). The rate of medication errors was 0.6%. The most common (80.6%) error was that the dispensed medicine dose was larger or smaller than indicated on the prescription. In addition, prescribing an FDC medicine increased the chance of making an error by 2.5 times on average. Regulatory changes dramatically affected the medicine-prescribing habits of INNs. The increase in FDC prescription rates may align with the recommendations of the 2018 ESC/ESH guidelines. The proportion of total errors is estimated as low, but control mechanisms are needed to prevent them.

## 1. Introduction

Standardisation of drug nomenclature is essential because a medicine may be sold by many different brand names, or a branded medication may contain more than one active substance. An international nonproprietary name (INN) plays a crucial role in identifying pharmaceutical substances or active pharmaceutical ingredients by a unique name that is globally recognised and is public property. A nonproprietary name is also known as a generic name [1].

The National Health Service (NHS) of the Republic of Latvia and an administrative agency of the Ministry of Health maintain medicine lists to be reimbursed for outpatient treatment. Depending on the diagnosis, there are three levels of reimbursement for pharmaceuticals: 100%, 75% or 50%. Drugs in the reimbursement lists are classified into three groups—A, B and C:list A, which includes medicines of equivalent effectiveness;list B, in which the medicines included do not have equivalent effectiveness with the medicine to be reimbursed;list C, which includes medicines in which the costs for the treatment of one patient exceed EUR 4268.62, and the manufacturer undertakes to cover the medication reimbursement costs for a particular number of patients with its own resources [2,3,4].

In addition, a separate list M was created, which includes medicines for pregnant women, women in the postnatal period up to 70 days with 25% reimbursement and children up to the age of 24 months with 50% reimbursement [4].

According to the Organisation for Economic Co-operation and Development (OECD) statistics, the proportion of costs spent on pharmaceuticals and medical devices in Latvia is the fifth highest in the EU. In order to promote the use of generics and reduce patient costs for medicine, INN use became a mandatory prescribing medicine within reimbursement medicine list A in Latvia on 1 April 2020 [4,5]. This means that physicians should prescribe medicines using the active ingredient name of the medicine, not its brand name. Simultaneously, pharmacies should dispense the lowest price medicine from the same therapeutic effect analogues to patients. The procedure was recommended by the WHO and OECD based on successful integration in Lithuania, Estonia, German, Finland, Denmark and other European countries [5,6].

Cardiovascular diseases are the leading cause of mortality [7]. In addition, adherence to medication among patients with arterial hypertension in Latvia was rated as low (45.9% nonadherence) [8,9]. Reducing the number of pills by FDC medicine to simplify the drug regimen is endorsed by ESC/ESH guidelines for arterial hypertension management. These combinations should contain drug classes such as angiotensin-converting enzyme inhibitors (ACEIs), angiotensin receptor blockers (ARBs), calcium channel blockers (CCBs) or diuretics, which are safe and effective in treating hypertension. In addition, beta-blockers (BABs) are considered at any treatment step with a specific indication of resistant hypertension, heart failure, angina, postmyocardial infarction, atrial fibrillation and pregnancy [7,10]. Because of their widespread use in practice and availability as a monocomponent and FDC medication on the market, bisoprolol or/and perindopril prescriptions were selected as the analysed study cohort in this study [11,12].

Globally, unsafe medication practices and medication errors are the leading causes of injury and avoidable harm [13]. Medication prescribing and dispensing are fundamental practice aspects, and related errors may be critical to patient safety [14]. A medication error (ME) is defined by the Europe Medicine agency as ‘an unintended failure in the drug treatment process that leads to, or has the potential to lead to, harm to the patient’. It was reported that the most common preventable cause of undesired adverse and major public health burden events are mistakes in prescribing, dispensing, storing, preparing and administering medicine [15]. However, electronic prescription databases were not previously used to quantitatively identify prescription-related errors in Latvia.

This study aimed to examine the prescribing and dispensing practice for bisoprolol or/and perindopril and their combination containing reimbursed medicine from list A after the regulatory changes for INN mandatory use. Moreover, evaluating the prevalence of medication errors and characterising them is significant for practice and patient safety. Understanding these issues may contribute to developing national policies and strategies for prescribing and dispensing and medication error identification, reporting and prevention.

## 2. Materials and Methods

### 2.1. Study Subjects and Data Collection

This retrospective study was performed using the Latvian NHS electronic prescription database. All bisoprolol or/and perindopril and their combination prescriptions based on ATC (C07AB07, C07FB07, C07BB07, C09BX02, C10BX15, C10BX11, C09AA04, C09BB04, C09BX01, C09BA04, C10BX14, C10BX13) for arterial hypertension (I10) with reimbursement from 1 April 2018 to 31 March 2019 and from 1 April 2020 to 31 March 2021 were analysed. Periods were chosen because of regulatory changes in prescribing reimbursed medicines as compulsory use of international nonproprietary names (INNs) since 1 April 2020.

All reimbursed paper prescriptions’ data were credited in the NHS prescription database after medicine dispensing at pharmacies, so all dispensed medicine prescriptions (paper and electronic) with financial compensation from the government were analysed. Prescribed with reimbursement but not purchased medicine prescriptions data were not obtained due to system limitations. All received records were anonymous and encrypted, which is why reversible identification was impossible. Each prescription record covered patients’ data (age, gender and residence area), information related to the prescribed drug (marketing authorisation number, brand name or INN, derivate INN for brand names, dose and information about physician’s speciality and workplace location) and dispensed drug (marketing authorisation number, brand name, derivate INN and dose) and the diagnosis for which the medication was prescribed.

### 2.2. Sample Size and Sampling Technique

Since the population size was large (1,353,100 records in the first period and 1,061,227 records in the second period), some variables contained ‘raw data’ that could not be cleaned automatically or manually without a lot of resources and time. Before it was further processed, cleaned and analysed, a random sample was selected according to the following parameters: confidence level = 99% and margin of error = 1%. A random number generator was used to ensure that each entry had the same chance of being selected. For further analysis, 16,539 prescriptions were randomly selected.

### 2.3. Reference Medication and Medication Error Identification

Medicine list A to be reimbursed updates every 3 months on 1 January, 1 April, 1 July and 1 October. All related information, including the brand name, ATC code, marketing authorisation number, drug form, dosage, package size, medicine full price and price with VAT, reference price, and price changed depending on compensation size, was available on the NHS website [16].

If a reference brand name according to actual reimbursement list A or INN was used for medicine prescribing, it was noted as ‘reference medicine’. Not all FDC medicines had reference medications in one reimbursement group. If the medication contained two or more active substances and did not have reference medication at a specific dose, the government reimbursed only the sum of the reference prices for active substance monocomponents. For example, the cheapest available medication on the market containing bisoprolol/perindopril 5 mg/5 mg 30 tablets costs EUR 10.23. However, the references in this group were not provided, and the 100% compensation size should be only EUR 3.48, not EUR 10.23, because the compensation price is based on monocomponent medication prices (bisoprolol 5 mg 30 tablets at EUR 0.88 plus perindopril 5 mg 30 tablets at EUR 2.60 equals EUR 3.48). In this study for this case, if the physician prescribed the cheapest brand-name medication or used INN for FDC medicine that did not have a reference in a group, it was marked as ‘the cheapest in the reimbursement group’.

Physicians can prescribe medicine with reimbursement by the government only from the Latvian National Health Service approved lists in specific drug forms and doses. For example, if the prescribed dose for an active substance or FDC was not available on the medicine lists to be reimbursed, it was evaluated as an error because the patient would not receive the same medicine with financial compensation at the pharmacy. According to Latvian regulation, pharmacists are prohibited from changing patients’ therapy. The primary duty is to dispense physicians’ prescribed medication and provide essential medical information exactly as prescribed, and deviations such as dispensing lower or higher doses are regarded by regulatory institutions as errors. The prescription should be rewritten in case of medicine unavailability in stock or patient preference or for any other reason why the exact prescribed medicine cannot be dispensed.

### 2.4. Data Analysis

A 95% confidence interval was calculated for the metrics to indicate the range likely to contain the true population parameter. A chi-square test of two proportions with continuity correction was conducted (if the assumption of adequate cell size was met) to determine whether there were existing differences in a dichotomous dependent variable between two groups of the independent variable. If the assumption of adequate cell size was violated, Fisher’s exact test was used. Linear regression was conducted to test the relationship between the continuous dependent variable and the continuous independent variable. Binomial regression was conducted to test the relationship between the dichotomous dependent variable and the dichotomous independent variable. *p*-values of less than 0.05 (*p* < 0.05) were considered significant. Data are presented as means ± SDs for normally distributed continuous variables and as medians and interquartile ranges for non-normally distributed continuous variables unless otherwise stated. Statistical data analysis was performed by applying the R 4.1.2. programme.

## 3. Results

### 3.1. Patients’ Demographic Data

In this subgroup of bisoprolol or/and perindopril prescriptions for arterial hypertension treatment, patients’ mean age was 68.3 ± 12.5 years. The ratio of prescriptions for women was higher than for men in both periods. More detailed information is summarised in Table 1.

### 3.2. Prescription Information Filled by Physicians

Medicines were prescribed most often (95.7%) by family doctors, followed by internists in 1.2% and cardiologists in 0.6% of cases. The proportions of INN and brand-name use in the two periods were significantly different: the proportion of INN prescriptions increased from 2.1% in the first period to 92.3% in the second period (*p* < 0.001, φ = 0.903). Nevertheless, during the second period within 1 year after the adoption of the regulation (from 1 April 2020), the proportion of brand-name medicine prescriptions significantly increased over time (F(1,10) = 7.38, *p* = 0.022, R^2^ = 0.425). These changes are shown in Figure 1.

Physicians prescribed FDC medicine containing bisoprolol or/and perindopril in 60.8% of prescriptions in the first period and in 66.5% in the second (*p* < 0.001, φ = 0.059). A significant difference in FDC medicine-prescribing frequency was recorded between younger (<80 years) and elderly patients (*p* < 0.001, φ = 0.097), in which monocomponent medications were prescribed in 33.9% v. 45.6% of cases. Nonetheless, FDC medicine use recommendations were more common than monocomponents for all age groups. The most popular (22.2%) FDC was ACEI with diuretics, namely perindopril/indapamide. Figure 2 shows a trend of a growing proportion of FDCs containing three active substances, which increased from 17.8% to 23.2%.

Medical practitioners commonly prescribing FDC medicine used brand names in 60.1% v. 76.5% of all brand-name prescriptions in both periods. Brand-name prescribed medicine in the majority was more expensive than reference medicine or the cheapest in the reimbursement group, but this proportion decreased by 16.0% after regulatory changes (Table 2). Contemporaneously, the proportion of prescribed medicine that did not have analogues in the reimbursement group increased from 14.7% to 23.0%.

### 3.3. Description of Prescribed Medicine Dispensing and Detected Errors

Only in 5.4% of all physicians’ brand-name medicine prescriptions did pharmacists or pharmacist assistants change and dispense another brand-name medicine. The rate of MEs was 0.6% (*n* = 98, 95% CI 0.48–0.72%). It was half as high in the second period compared with the first at 0.81% (95% CI 0.63–1.03%) v. 0.39% (95% CI 0.25–0.53%), respectively. The most common (*n* = 79, 80.6%) ME was that the dispensed medicine dose was larger or smaller than indicated in the prescription. In 6 (6.1%) cases, we were unable to identify prescribed doses; in 10 (10.2%) cases, the physician indicated doses incorrectly; in 1 (1.0%) case, a drug from another pharmacological group was dispensed; and in 2 (2.0%) cases, the number of active substances of FDC medicine was changed. In 67 of 98 (68.4%) total error cases, INNs were used to prescribe medicine, of which 55 were FDC medicine. FDC prescribing significantly predicted the chance to make an error (B = 0.935, z = 3.58, *p* < 0.001, OR = 2.5). This suggests that the physicians who prescribed FDC increased their chance of making an error by 2.5 times on average (95% CI 1.52–4.25).

## 4. Discussion

INNs play a significant role in the pharmaceutical industry, minimising confusion between various available brand names. Regardless, brand-name use was excessively prevalent in practice, showing this study’s results. Professionals are not isolated from marketing, market changes, medicine prices and other factors influencing their medication choice preferences. At the same time, writing a short brand name is faster and easier than writing a two–three-word-long INN for an FDC medicine prescription. This situation reversed categorically when the recommended use of the INN became mandatory after the 1 April 2020 regulation changes, which was why INN use increased by 90.2%. Since the brand-name use trend increased but did not exceed the proportion of 10%, in these circumstances, it was noticed that recommendations were insufficient. Each innovation should become mandatory if the government is interested in integrating it into daily practice. Using a brand name for medicine, physicians prescribed and precisely recommended the specific manufacturer medication, which was not always the cheapest available on the market, and pharmacists, in most cases, dispensed to patients exactly this product. According to the Latvian medicine reimbursement system, the difference between the reference or the cheapest medication in the group and a specific brand-name medication is paid by the patient, which is an important financial factor, especially for seniors. After the regulation changes to reduce patients’ expenses for medicine, INN use indicated that patients would receive the cheapest therapeutic analogue available at a pharmacy without copayment ability for medication preferences or if the price difference was only EUR 0.01. This period was difficult for patients, especially for long-term therapy users, because many patients who previously used specific brand-name medications did not have an increased reimbursement for those brands. Practice shows that such a mechanism increases the competitiveness of manufacturers to provide lower prices and reduces the cost of medicines to the population [3,16]. Still, there are no available data on how many patients refused dispensed reimbursement medicines to purchase specific medication for full price.

A limitation of the study was that the medicine dispensing date was not provided, so it was impossible to identify the actual medicine price for patients at pharmacy visits due to the updating of the reimbursement list at least every 3 months. For this reason, pharmacists’ duty to dispense references or the cheapest medicine for patients was not evaluated. Another limitation was that we may not have accounted for all factors that might have influenced prescription habits between the two periods, but in our practice, the medical community witnessed a surge in INN prescriptions since 1 April 2020, when the new regulation came into effect. No other significant legislation changes regulating prescriptions took place within 2 years before April 2020. The calls from the Ministry of Health to physicians to prescribe INNs and to raise public awareness among patients about generic alternatives had been conducted well before April 2018 but without apparent success. Therefore, in our opinion, the first period represents the state of prescription habits before the new regulation. The study’s strength describes how world institutions’ recommendations, such as the WHO and the OECD, and their implementation mechanisms can impact daily medicine-prescribing habits and the dispensing process at the state level.

The ESC/ESH guideline recommendations on pharmacotherapy in hypertension recommend FDC drugs to simplify the regimen and improve adherence and blood pressure outcomes compared with monocomponent pill combinations [7]. While low (10.9%) FDC medication prescribing was observed in Germany, this study demonstrated relatively high (66.5%) FDC drug use for arterial hypertension treatment, conceivably because Latvia has a relatively high penetration rate of generics in the market [17,18]. Moreover, more varying dosages and new combinations have become available, for example, FDC with HMG-CoA reductase inhibitors.

The interest in reference medicine led to the situation that every time the reimbursement list was updated (at least every 3 months), there was a different manufacturer of the cheapest analogue of the reference medicine, not only among monocomponent or FDC medicine but also in each dose group. Physicians were not sure which medication brand patients would obtain at a pharmacy. Patients were worried that they received the same medicine in different-looking packages, which may decrease patient satisfaction with therapy and lead to lower medication adherence levels. The pattern corresponds to survey results that patients, especially seniors, strongly preferred brand-name medications recommended by physicians and, therefore, were not ready to accept generics [18]. Pharmacists did follow each reference medicine existing in stock and drug wholesalers. If reference medicines were unavailable, pharmacists would dispense the next cheapest. It was difficult to predict the volume of medicines purchased for pharmacy stock. After all, if the reference medications are changed, this medication will not be allowed to dispense in the next period of the reimbursement system because it has become more expensive than the new reference. Due to the necessity to dispense reference medicine to reimburse prescriptions every day, large accumulations of medications that were not possible to return were formed at pharmacies and drug wholesalers. The pandemic also had a negative impact on the supply of medicines when there was no clarity about the availability and supply of drugs [19,20].

The overall rate of MEs was 0.6%. The error rate was lower than in the UAE (2.6%), Trinidad and Tobago (2.1%) and the USA (1.7%) but higher than in Denmark (1/10,000 cases) [21,22,23,24]. However, a statistical comparison with other studies’ results is not possible due to different methodology approaches, analysis of data types, varied healthcare professionals’ duties among countries and other factors. The most common mistake was changed dosages, which was most likely due to pharmacists’ heavy workload and resemblances among medications. It is useful to note that only electronic prescriptions in the reimbursement system were analysed in this study. The real ME value among all prescriptions (including full-price medicine prescriptions) could be higher based on the results from other electronic record study in Latvia [25]. Moreover, health authorities in Latvia focus on controlling the dispensing of reimbursement system medications, with much less oversight of over-the-counter and ordinary (full-price) prescription medications.

INN use for prescribing medicine is a good practice, so health authorities should discuss further mandatory INN use integration for each prescription in Latvia. It is important to develop a system that does not limit patient preferences for medication choices by regulation. By law, a physician has the right to use brand names for prescribing no more than 30% of practice prescriptions, mentioning replacement inability reasons. Altogether, patients who want specific brand-name medications and are ready to overpay are also entitled to financial compensation for medicine in the reimbursement system. The government reimbursement cost will not increase, but patients will have a larger medication selection depending on their choice. Potentially, it will be valuable to update the reimbursement medicine lists semiannually or notify all pharmaceutical supply chain participants in advance about reference changes so that there is no stagnation of medicines and associated financial losses.

## 5. Conclusions

The example of prescribing and dispensing bisoprolol or/and perindopril suggests that INN mandatory use for reimbursement prescriptions significantly influenced physicians’ prescribing habits. The proportion of INN prescriptions increased by 90.2%. FDC medicine for arterial hypertension treatment was prescribed in 60.8% of cases in the first period and in 66.5% of cases in the second and rated as a relatively high index with an increasing tendency to recommend three-component medicines. In general, FDC medicine prescribing was more favoured, which aligned with the ESC/ESH guideline recommendations. At dispensing, only in 5.4% of all brand-name medicine prescriptions did pharmacists or pharmacist assistants change and dispense another brand-name medicine. Despite relatively low (0.6%) MEs, more mechanisms or practical solutions should be integrated into daily practice to reduce the risks associated with drug safety. As a downside, INN use for FDC medicine prescribing was associated with a higher ME risk.

## Figures and Tables

**Figure 1 ijerph-19-10156-f001:**
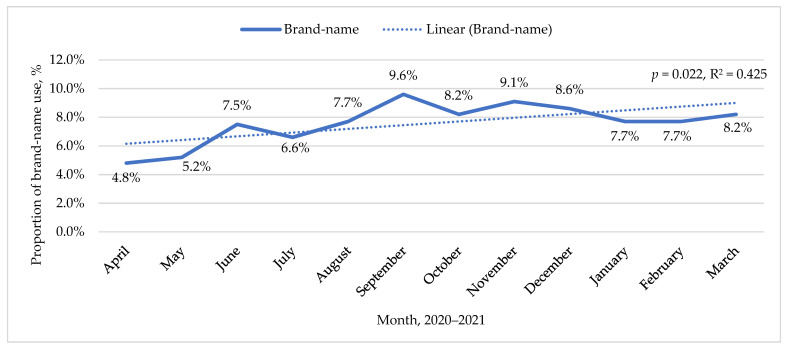
Changes in the proportion of brand-name use after 1 April 2020.

**Figure 2 ijerph-19-10156-f002:**
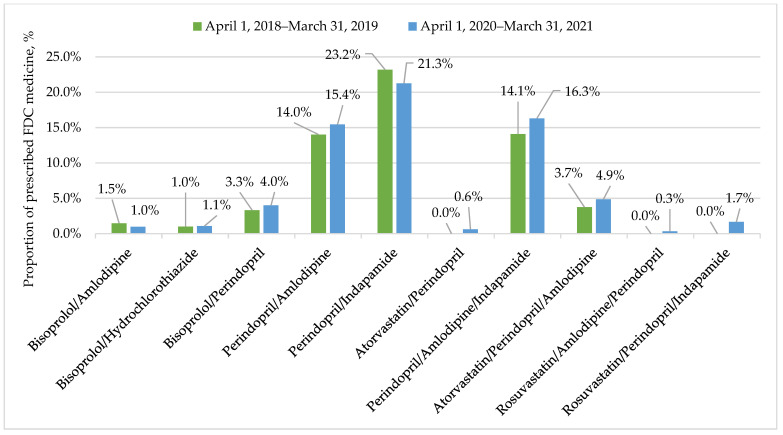
Description of prescribed FDC medicine for arterial hypertension treatment in each period.

**Table 1 ijerph-19-10156-t001:** Characteristics of patients.

Variable	Period	
	1 April 2018–31 March 2019	1 April 2020–31 March 2021
Total number of prescriptions, *n*	8279	8260
Number of patients, *n*	7876	7912
Age, years (mean ± SD)	68.6 (12.4)	68.0 (12.6)
Female sex, *n* (%)	5102 (64.8)	4949 (62.6)
Residence area *, *n* (%):		
City	3845 (48.8)	4275 (54.0)
Countryside	4024 (51.1)	3626 (45.8)

* Missing data were not included.

**Table 2 ijerph-19-10156-t002:** Prescription classification by physicians’ used medicine name and medication status in the reimbursement group.

Variables	Period	
	1 April 2018–31 March 2019	1 April 2020–31 March 2021
Total number of prescriptions, *n*	8279	8260
INN use, *n* (%)	175 (2.1)	7625 (92.3)
Brand-name use, *n* (%)	8104 (97.9)	635 (7.7)
Reference medicine or the cheapest in the reimbursement group *, *n* (%)	1087 (13.4)	134 (21.1)
More expensive than reference medicine or the cheapest in the reimbursement group *, *n* (%)	5829 (71.9)	355 (55.9)
Do not have analogues *, *n* (%)	1188 (14.7)	146 (23.0)

* Medication status in the reimbursement group among brand-name medicine prescriptions.

## Data Availability

Not applicable.

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
