# Peer review of "The Impact of International Nonproprietary Names Integration on Prescribing Reimbursement Medicines for Arterial Hypertension and Analysis of Medication Errors in Latvia"

_ijerph, 2022, doi:10.3390/ijerph191610156_

Round 1

Reviewer 1 Report

The study has a high merit on cross-referencing prescription and dispensing databases! But without extra knowledge on what happened between both stages, conclusions should only be made with lots of caution.

Any deviation from prescription to dispensing is categorized as a Medication Error : I do not agree, as intelligence is missing on the reasons, that can be legitimate : prescriber made a mistake, medicine unavailability and substitution in the interest of therapy continuity, patient preference, etc. The authors should describe this.

The comparison of prescribing and dispensing characteristics in both periods should at least take into account the possible differences in marketing and availability of the concerned products on the Latvian market. Present comparison seems to hypothesize that these are equal in both periods, I doubt that.

April 2020 (start second period) equaled lockdowns all over Europe, I suppose in Latvia as well. We see relative prescription numbers, but what about absolute numbers? Other studies have seen steep decline in March/April/May 2020.

Authors state that INN prescribing for FDC is more cumbersome that brand name prescribing (line 208 - I tend to agree), but at the same time they state that compulsory INN prescribing has risen the proportion of FDC prescription. The authors should at least try to explain the phenomenon. Other factors might play a role. 

The variable "residence area" of the patient (line 102 and table 1) is presented but not discussed. I don't think it's of any use.

It is unclear of prescription software had changed between the two periods. The authors should at least describe how electronic prescriptions are made (e.g. from alphabetical drop down list, by INN, by ATC, etc)

Line 62 : an interesting statement on adherence to hypertension medication. How was it influenced by compulsory INN prescribing and compulsory dispensing of the cheapest medicine from the list ?

English needs improvement : sometimes verb is missing (line 16, 109,...) or doubled (line 76,...)

Medicines in scope should be described by ATC

Author Response

Dear Reviewer,

Thank you for comments. Please find below our response to the comments received as well as actions taken to accommodate changes to the manuscript.

Should you have any queries, please do not hesitate to contact us.

  1. The study has a high merit on cross-referencing prescription and dispensing databases! But without extra knowledge on what happened between both stages, conclusions should only be made with lots of caution.

The authors agree that we may not account for all factors that may have influenced prescription habits between the two periods, but in our practice the medical community witnessed a surge of INN prescriptions since April 1, 2020 when the new regulation came into effect. No other significant legislation changes regulating prescription took place within two years before April 2020. Also the calls from the Ministry of Health to physicians to prescribe INN and raising public awareness of patients of generic alternatives had been conducted well before April 2018, but without apparent success. Therefore, in our opinion the first period represents the state of prescription habits before the new regulation.

In our view the note of caution was implied with the word “suggests” in our conclusion “The example of prescribing and dispensing bisoprolol or/and perindopril suggests that INN mandatory use for reimbursement prescriptions significantly influenced physicians prescribing habits”.

We have added these limitations of the study in the discussion which we hope will address the reviewer’s concerns.

  1. Any deviation from prescription to dispensing is categorised as a Medication Error : I do not agree, as intelligence is missing on the reasons, that can be legitimate : prescriber made a mistake, medicine unavailability and substitution in the interest of therapy continuity, patient preference, etc. The authors should describe this.

We agree that not all inconsistencies defined in our manuscript as medical errors may be explained as true errors in clinical practice. This study is based on medical error evaluation from a regulatory standpoint. The medical error identification process, now more detailed, is described in the Materials and Methods part.

  1. The comparison of prescribing and dispensing characteristics in both periods should at least take into account the possible differences in marketing and availability of the concerned products on the Latvian market. Present comparison seems to hypothesise that these are equal in both periods, I doubt that.

Official statistics of medication availability and drug marketing description were not published. Based on our available bisoprolol and perindopril and their combinations prescriptions data, we calculated that 14.4% of all stage I prescriptions did not have analogues compared to 23.3% in the stage II. It is due to new reimbursement groups not existing in the first period. Excluding those new medicine groups, only 19.4% of prescribed medicine did not have analogues in the second period. The increase of 5.0% was due to raised FDC popularity that, in most cases, did not have analogues among reimbursement List A. All other medicines had analogues, and specific manufacture limited availability cannot affect the dispensing process in this study.

We agree that marketing changes can be confounding factors. However, this information cannot be quantified, which we marked as a study limitation too.

  1. April 2020 (start second period) equaled lockdowns all over Europe, I suppose in Latvia as well. We see relative prescription numbers, but what about absolute numbers? Other studies have seen steep decline in March/April/May 2020.

The government declared a state of emergency on March 13, 2020, with many epidemiological safety measures and restrictions. Also, we have data that starts only on April 1, 2020. COVID-19 pandemic possibly limited paper prescription circulation, restricting healthcare provided in person and influencing physicians' habit of prescribing more electronic prescriptions. Electronic and paper prescription ratio was not provided from the database, so we focused more on prescription content analysis.

This hypothesis is interesting for future study investigation.

  1. Authors state that INN prescribing for FDC is more cumbersome that brand name prescribing (line 208 - I tend to agree), but at the same time they state that compulsory INN prescribing has risen the proportion of FDC prescription. The authors should at least try to explain the phenomenon. Other factors might play a role.

With all due respect, we are not sure we understood the question and to which line in the text the reviewer refers to regarding “but at the same time they state that compulsory INN prescribing has risen the proportion of FDC prescription”. The INN prescribing became mandatory as of April 1, 2020 for all state-reimbursed drugs in A list which included both monotherapies and FDCs. We do not state that proportion of FDCs increased due to INN integration. It likely increased in accordance with ESC/ESH recommendations and new FDC combination availability in the market (the third paragraph in discussion). In the manuscript it was mentioned that INN use for FDC medicine prescribing was associated with higher ME risk.

We hope we have answered the question.

  1. The variable "residence area" of the patient (line 102 and table 1) is presented but not discussed. I don't think it's of any use.

We would like to keep "residence area" as it is one of the patients' characteristics that was provided in prescriptions. However, significant differences between patients living in cities and the countryside were not detected.

  1. It is unclear of prescription software had changed between the two periods. The authors should at least describe how electronic prescriptions are made (e.g. from alphabetical drop down list, by INN, by ATC, etc)

We have the e-Health (E-veselība) platform that collects prescription data and stores all prescription-related information in the database. Prescribers can work directly with the E-Health platform or via an intermediary program. Each specialist can connect to E-Health anytime from the computer as required. Pharmacies also work with different intermediary programs to make the dispensing process easier and faster. Each program may have its own specifics but meets the standards of E-Health requirements for data exchange. In both periods, the system operating processes of the e-Health were the same.

Latvian NHS provided described data selection (2.1. Study Subjects and Data Collection).

  1. Line 62 : an interesting statement on adherence to hypertension medication. How was it influenced by compulsory INN prescribing and compulsory dispensing of the cheapest medicine from the list ?

We absolutely share the concerns of the reviewer that these frequent changes of the names of antihypertensive agents may affect patients’ adherence. Patients in Latvia often trust physicians' choice of the specific brand or they have used to a specific drugs with recognizable names. After INN integration, prescribers were not allowed to use brand-names, so patients, in our observation, were not satisfied with the new manufacturer's reference brand (the second paragraph in the discussion) and frequently were confused about “changes” of the drugs.

It would be interesting to study adherence levels in more detail after the legislation changes in a separate study.

  1. English needs improvement : sometimes verb is missing (line 16, 109,...) or doubled (line 76,...)

We hope the corrections now meet the requirements.

  1. Medicines in scope should be described by ATC.

Information according to ATC was added to the text.

Reviewer 2 Report

Dear authors, 

I would like to add some suggestions for improving your manuscript:

Abstract

Please rewrite the introduction and aims, It is not very clear

Introduction and overall

Please place reference number [“] before the.

Table 2 missing parenthesis

3.0)

Discussion

Please add the limitations and strengths of the manuscript

Author Response

Dear Reviewer,

Thanks for your suggestions. All your comments were added to the manuscript.

Round 2

Reviewer 1 Report

Thank you for adequately responding to the identified shortcomings, it greatly improved the quality of your work.